# Astaxanthin Added during Post-Warm Recovery Mitigated Oxidative Stress in Bovine Vitrified Oocytes and Improved Quality of Resulting Blastocysts

**DOI:** 10.3390/antiox13050556

**Published:** 2024-04-30

**Authors:** Linda Dujíčková, Lucia Olexiková, Alexander V. Makarevich, Alexandra Rosenbaum Bartková, Lucie Němcová, Peter Chrenek, František Strejček

**Affiliations:** 1Research Institute for Animal Production Nitra, National Agricultural and Food Centre (NPPC), Hlohovecká 2, 951 41 Lužianky, Slovakia; linda.dujickova@nppc.sk (L.D.); lucia.olexikova@nppc.sk (L.O.); alexander.makarevic@nppc.sk (A.V.M.); peter.chrenek@nppc.sk (P.C.); 2Department of Botany and Genetics, Faculty of Natural Sciences and Informatics, Constantine the Philosopher University in Nitra, Tr. A. Hlinku 1, 949 01 Nitra, Slovakia; arbartkova@ukf.sk; 3Laboratory of Developmental Biology, Institute for Animal Physiology, Genetics of the Czech Academy of Sciences, Rumburská 89, 277 21 Liběchov, Czech Republic; nemcova@iapg.cas.cz; 4Institute of Biotechnology, Faculty of Biotechnology and Food Science, Slovak Agricultural University in Nitra, Tr. A. Hlinku 2, 949 76 Nitra, Slovakia

**Keywords:** vitrification, oocytes, astaxanthin, embryo development

## Abstract

Various antioxidants are tested to improve the viability and development of cryopreserved oocytes, due to their known positive health effects. The aim of this study was to find whether astaxanthin (AX), a xanthophyll carotenoid, could mitigate deteriorations that occurred during the vitrification/warming process in bovine oocytes. Astaxanthin (2.5 µM) was added to the maturation medium during the post-warm recovery period of vitrified oocytes for 3 h. Afterward, the oocytes were fertilized in vitro using frozen bull semen and presumptive zygotes were cultured in the B2 Menezo medium in a co-culture with BRL-1 cells at 38.5 °C and 5% CO_2_ until the blastocyst stage. AX addition significantly reduced ROS formation, lipid peroxidation, and lysosomal activity, while increasing mitochondrial activity in vitrified oocytes. Although the effect of AX on embryo development was not observed, it stimulated cell proliferation in the blastocysts derived from vitrified oocytes and improved their quality by upregulation or downregulation of some genes related to apoptosis (*BCL2*, *CAS9*), oxidative stress (*GPX4*, *CDX2*), and development (*GJB5*) compared to the vitrified group without AX. Therefore, the antioxidant properties of astaxanthin even during short exposure to bovine vitrified/warmed oocytes resulted in improved blastocyst quality comparable to those from fresh oocytes.

## 1. Introduction

Cryopreservation offers long-term storage of genetic material for several decades. It is widely used in assisted reproduction in humans, gamete preservation of endangered animal species, or in farm animals, where cryopreservation might provide the storage of gametes from genetically valuable individuals, with the possibility of increasing their offspring numbers. However, oocyte viability and developmental potential after warming remain low and variable worldwide.

It has been found that vitrification/warming can cause elevated reactive oxygen species (ROS) production [1], lipid peroxidation [2], abnormal meiotic spindle [3], pre-mature cortical granule release, and zona pellucida hardening [4], as well as other damage on organelles, leading to decreased developmental potential. Optimization of methodology by choosing the right combination and concentration of cryoprotectants, culture condition, and media composition could reduce the negative impact of vitrification on the oocytes. Certain damage, caused by the vitrification process itself, cannot be completely removed, and it requires at least mitigating the external influence of the whole in vitro process.

The above-mentioned increased ROS formation causes an imbalance between the formation and destruction of ROS, also called oxidative stress [5]. Free radicals are normally derived from metabolic processes in the body. However, some external influences, such as in vitro conditions or vitrification/warming, can lead to overproduction of free radicals, which have a negative impact on the oocyte and subsequent embryo development [6]. This is the case when antioxidants come in. These are molecules which can decrease oxidative damage either directly, by reacting with free radicals, or indirectly, by inhibiting the activity of free radical-generating enzymes, or by enhancing the activity of intracellular antioxidant enzymes [7]. Antioxidants can act as free radical scavengers and, therefore, protect cells against potential damage by repairing damage caused by ROS and RNS (reactive nitrogen species) [8]. Exogenous and endogenous antioxidants can offset extreme ROS concentrations generated from vitrification/warming and improve embryo quality [9]. However, the endogenous antioxidant system is insufficient in vitro; therefore, adding exogenous antioxidants could be an option to improve in vitro culture conditions [10]. Several authors in their studies [11,12,13,14] used antioxidants in maturation, vitrification, or culture media to increase cryotolerance and viability of oocytes after vitrification and subsequent embryo development. In recent years, there has been a significant interest in testing bioactive molecules of marine origin, as they showed high potential for health benefits.

Astaxanthin (AX; 3.3′-dihydroxy-b-b′-carotene-4,4′-dione) is a red pigment naturally contained in algae, yeast, salmon, trout, krill, shrimp, and crayfish. Astaxanthin belongs to the family of xanthophylls, the oxygenated derivates of carotenoids, whose synthesis in plants derives from lycopene [15]. It is well-known as a highly potent natural antioxidant, apparently because of its structure, containing two hydroxyl groups and two carbonyl groups in the b-rings [16]. Several non-reproductive effects of astaxanthin, such as protective effects on the cardiovascular system [17], anti-inflammatory [18] and anti-cancer [19,20] effects, were reported. Dietary supplementation of astaxanthin suppressed oxidative stress [21,22] and improved immune response and growth [23] in aquatic species. The importance of astaxanthin has been proved also in marine animal reproduction [24,25].

In the field of farm animal reproduction, the effects of astaxanthin on improving the quality of porcine oocytes after aging in vitro [26] and vitrification/warming [27], development of bovine cloned animals [28], or cryoprotective properties on boar sperm [29] are already known. However, the effect of AX especially on bovine vitrified oocytes has not been documented. The aim of our study was to evaluate the effect of AX added during post-warm recovery on bovine vitrified oocytes and their developmental capacity.

## 2. Materials and Methods

All chemicals were purchased from Sigma Aldrich (Saint-Louis, MO, USA) unless otherwise specified.

### 2.1. Oocyte Collection and In Vitro Maturation

Oocytes were aspirated from follicles (2–8 mm) of slaughtered-derived cow ovaries from a local abattoir by a 5 mL syringe with an 18 G needle. Only cumulus–oocyte complexes with homogeneous ooplasm and intact cumulus cell layers were selected for in vitro maturation (IVM; M199, 10% fetal bovine serum (FBS), 0.25 mM sodium pyruvate, 50 µg/mL gentamicin, 1 I.U. FSH/LH (Pluset)) at 38.5 °C and 5% CO_2_. The length of maturation lasted 21 h in the case of a vitrified group and 24 h in the case of fresh oocytes (control group; CONT).

### 2.2. Oocyte Vitrification and Warming

The vitrification/warming procedure was performed as described in our earlier work [30]. Briefly, oocytes were, after the in vitro maturation procedure, incubated for 12 min in the equilibration medium (M199, 3% ethylene glycol, 10% FBS, 50 µg/mL gentamicin). Afterward, the oocytes were washed in the vitrification medium (M199, 10% FBS, 30% ethylene glycol, 1 M sucrose) for 25 s, placed onto electron microscopy grids (EMGs), immediately plunged into liquid nitrogen, and stored in a container with liquid nitrogen for at least one week. Afterward, oocytes were warmed in a warming medium (M199, 10% FBS, 0.5 M sucrose, 50 μg/mL gentamicin) for 1 min at 37 °C, washed in a set of media with decreasing sucrose concentrations (0.25; 0.125; 0.0625 M), and finally washed in a medium without sucrose (M199, 10% FBS, 50 μg/mL gentamicin). Post-warm oocytes were transferred to the maturation medium either with 2.5 μM astaxanthin (VIT-AX) or without the AX addition (0 μM; VIT) for a 3 h recovery period.

### 2.3. Detection of ROS Formation in Oocytes

The intracellular ROS was assessed by fluorescent staining using fluorescent reagent CellROX™ Green (Invitrogen, Waltham, MA, USA). Denuded oocytes were incubated in a dye mixture (according to the manufacturer) for 30 min at 37 °C, washed in PBS-PVP solution (phosphate-buffered saline with 0.6% of polyvinylpyrrolidone), fixed in 4% formalin for 10 min, and mounted on a glass slide. Stained oocytes were scanned by an LSM 700 Zeiss confocal laser scanning microscope, and the ROS formation level was evaluated using ImageJ 1.53a software.

### 2.4. Detection of Lipid Peroxidation in Oocytes

Lipid peroxidation in oocytes was assessed by fluorescent staining using BODIPY™ fluorescent reagent (Invitrogen, Waltham, MA, USA). Denuded oocytes were incubated in a drop of PBS-PVP with BODIPY™ (10 μM) under the mineral oil at 37 °C and 5% CO_2_ for 30 min. Afterward, oocytes were washed in PBS-PVP three times and mounted on a glass slide. Stained oocytes were then scanned by an LSM 700 Zeiss confocal laser scanning microscope (Carl Zeiss Slovakia, s.r.o., Bratislava, Slovak Republic) and the lipid peroxidation level was evaluated using ImageJ software.

### 2.5. Analysis of Mitochondrial and Lysosomal Activity in Oocytes

The oocytes were simultaneously stained for mitochondria, by fluorescent reagents MitoTracker^®^ Green FM (Molecular Probes, Eugene, OR, USA), and for lysosomes, by LysoTracker™ Deep Red (Invitrogen, Waltham, MA, USA), to evaluate their status and activities. Briefly, 30–50 oocytes were incubated in 200 nM of MitoTracker^®^ Green for 30 min at 37 °C, washed in PBS-PVP three times, and then incubated in 50 nM of LysoTracker™ Deep Red for 30 min at 37 °C. Afterward, oocytes were washed in PBS-PVP, fixed in 4% formalin for 20 min, and mounted on a glass slide. Stained oocytes were scanned by an LSM 700 Zeiss confocal laser scanning microscope, and the mitochondrial and lysosomal activities were evaluated using ImageJ software.

### 2.6. In Vitro Fertilization (IVF) and Embryo Culture

Oocytes selected for IVF were placed under a mineral oil into fertilization drops of IVF-TALP medium (Tyrode-Albumin-Lactate-Pyruvate solution containing 6 mg/mL BSA, 10 µg/mL heparin and 50 µg/mL gentamicin), to which PHE (20 µM penicilamine, 10 µM hypotaurine and 1 µM epinephrine) solution and sperm suspension (2 × 10^6^/mL) were added. After 18 h of IVF procedure, presumed zygotes were cleaned of excessive cumulus cells and placed into a previously prepared in vitro culture dish containing B2 Menezo medium (with 10% FBS, 10 mg/mL BSA, 31.25 mM sodium bicarbonate and 50 μg/mL gentamicin) onto a monolayer of BRL (Buffalo Rat Liver; ECACC) cells for co-culture at 5% CO_2_ and 38.5 °C. The embryo cleavage rate was determined on day 2 and development to the blastocyst stage was monitored on day 8 of in vitro culture.

### 2.7. Fluorescent Staining of Blastocysts

The total cell number was evaluated by labeling the blastocyst nuclei using a Vectashield anti-fade medium containing DAPI fluorochrome (Vector laboratories, Burlingame, CA, USA). The occurrence of apoptotic cells was assessed by a TUNEL assay (Promega, Madison, WI, USA). Actin cytoskeleton was fluorescently stained using phalloidin-TRITC (Merck, Darmstatd, Germany). Briefly, selected blastocysts were fixed in a neutrally buffered formalin (4%) for 15 min and washed 3 times in PBS-PVP solution. For the TUNEL assay, blastocysts were permeabilized in 0.5% Triton X-100 solution in PBS for 30 min at room temperature and then incubated in a TUNEL mixture (according to the manufacturer) in a humidified chamber at 37 °C for 60 min. Afterward, the embryos were washed in PBS-PVP solution and incubated in phalloidin-TRITC for 30–60 min for actin cytoskeleton staining. Stained blastocysts were placed onto coverslips, covered with 2 μL of Vectashield with DAPI, attached to microslides to make a sandwich, and analyzed under a Leica fluorescence microscope. The actin cytoskeleton was qualified as either intact or damaged, based on the appearance of actin filaments in blastocysts, as described earlier [31]. In particular, intact actin was characterized by sharp staining of actin filaments in cell borders creating a clearly visible network, while damaged actin was characterized by lacking actin staining or actin filaments, mainly visible as aggregated clumps.

### 2.8. RNA Isolation and RT-qPCR

Determination of the gene expression was based on the analysis of RNA isolated from the blastocysts of the three tested groups (CONT, VIT, and VIT-AX). Total RNA was extracted using the RNeasy Micro kit (Qiagen, Hilden, Germany) and treated with DNAse according to the manufacturer’s instructions. The primers, which were designed using Beacon Designer v. 8.21 (Premier Biosoft, CA, USA), are listed in Table 1.

The Qiagen OneStep RT-qPCR Kit (Qiagen, Hilden, Germany) was used for one-step RT-qPCR analysis on a RotorGene 3000 cycler (Corbett Research, Sydney, Australia). RT-qPCR reaction conditions were the following: reverse transcription (50 °C, 30 min) followed by qPCR—initial activation (95 °C, 15 min), cycling (40 cycles), denaturation (94 °C, 20 s), annealing (determined at specific temperatures for selected primers with a duration of 20 s), and polymerization (72 °C, 30 s). Fluorescence detection was performed at a temperature 2–3 °C lower than the melting temperature of the selected genes to distinguish the presence of possible dimers. To confirm the specificity of the acquired products, melting curve analysis and gel electrophoresis (using a 1.5% agarose gel) with MIDORI Green Direct staining were performed. For determining the relative concentration of templates and performing gene expression analyses, the internal Rotor-Gene V6.01 comparative analysis software was applied. This was carried out by normalizing the abundance of *H2AFZ* as a reference gene.

### 2.9. Statistical Analysis

All data were analyzed using GraphPad Prism 8 (GraphPad Software, Boston, MA, USA). Oocytes stained with fluorescent reagents were evaluated using ImageJ software. The corrected total cell fluorescence (CTCF = integrated density—(area of selected cell x mean fluorescence of background readings) formula was used for fluorescence intensity determination. Values of relative fluorescence intensity were converted to arbitrary units (AU) and reported as the mean ± SEM (standard error of the mean). The differences in the mean value of the tested groups versus control group were evaluated by t-test. Embryo culture experiments were performed in five replicates. Development (cleavage and blastocyst rates), distribution according to cell number, and actin cytoskeleton were analyzed by Pearson’s Chi-square test with Yates’s correction for continuity. The incidence of apoptotic cells was represented as a percentage of the total cell number. The relative expression of tested genes was presented as the mean ± SD (standard deviation). The differences in the mean value of the tested groups versus the control were evaluated by the t-test. Values were considered as statistically significant at *p* < 0.05.

## 3. Results

### 3.1. Effect of AX Addition on Oocytes Post-Warming

#### 3.1.1. Mitochondrial and Lysosomal Status in Oocytes

Based on the relative fluorescence intensity (Figure 1), vitrification resulted in increased lysosomal activity (1.90 AU; *n* = 36) compared to the control (1 AU; *n* = 27) and AX group (0.85 AU; *n* = 30), while mitochondrial activity (1.31 AU) remained comparable with control (1 AU) and AX group (1.59 AU). Interestingly, mitochondrial activity in the AX group was significantly higher than in the control. The distribution pattern of mitochondria and lysosomes was mostly scattered throughout the ooplasm in all tested groups.

#### 3.1.2. ROS Production and Lipid Peroxidation in Vitrified/Warmed Oocytes

Based on the relative fluorescence intensity (Figure 2), vitrification significantly increased ROS production (3.69 AU; *n*= 31) compared to the control group (1 AU; *n*= 37), while AX suppressed ROS production in warmed oocytes (2.08 AU; *n*= 35). Increased ROS formation induced lipid peroxidation in warmed oocytes (1.83 AU; *n* = 55) based on relative fluorescence intensity using BODIPY™ reagent. Astaxanthin suppressed lipid peroxidation level (0.66 AU; *n* = 53) in vitrified/warmed oocytes on the control level (1 AU; *n* = 59).

### 3.2. Effect of Vitrification and AX on Embryo Development and Blastocyst Quality

A total of 484 (VIT—*n* = 258; VIT-AX—*n* = 226) warmed oocytes were selected for IVF, which represents 72.80% viability of oocytes after vitrification. The control group consisted of 201 fresh IVM oocytes. The cleavage rate was slightly lower in the vitrified group (58.14%; Table 2) compared to the control group (66.17%), while AX addition did not improve this value (54.87%). Similarly, the rate of blastocysts developed from vitrified oocytes (14.34%) was significantly lower compared to the control group (29.85%), and astaxanthin only slightly improved this parameter (17.26%).

The total cell number of resulting blastocysts was significantly lower in the vitrified group (94.03 ± 5.08), compared to the control (103.80 ± 2.81) group. However, AX significantly increased the total cell number (105.28 ± 4.45) compared to the vitrified group without AX (94.03 ± 5.08), thus demonstrating its proliferative effect on vitrified/warmed oocytes. The incidence of apoptotic cells (about 10%) was affected neither by vitrification nor by the AX addition (Table 2).

The ranging of obtained blastocysts according to their cell number was carried out to determine the proliferative effect of astaxanthin (Figure 3). Most blastocysts in the control (52.44%) and vitrified AX-supplemented group (55.56%) had a cell number in the range of 81–120 cells. On the contrary, most blastocysts with a lower number of cells (less than 80) occurred in the vitrified group (46.88%), while in the control (21.95%) and vitrified AX-supplemented (19.44%) groups, such blastocysts had the lowest ratio. Percentages of blastocysts with more than 120 cells were almost similar in the control group (25.61%) and in the vitrified AX-supplemented group (25.00%) and only slightly lower (21.88%) in the vitrified group. Therefore, these results provide evidence that vitrified oocytes significantly differ from control and AX-supplemented vitrified oocytes.

More than 70% of embryos, both in the control and vitrified group, had the intact structure of actin filaments (Table 3). Astaxanthin addition increased the rate of blastocysts with intact actin (82.76%) compared to the control (70.73%) and vitrified groups (70.83%) but the difference was not statistically significant. These observations suggest a generally stimulating trend of AX on actin cytoskeleton in blastocysts irrespective of vitrification.

### 3.3. Effect of Vitrification and AX on the mRNA Expression of Chosen Genes

The mRNA expression of selected genes referring to apoptosis (*BAX*, *BCL2*, *CAS3*, and *CAS9*), oxidative stress (*CAT*, *GPX4*, and *SOD2*), and developmental competence (*CDX2* and *GJB5*) in blastocysts was determined by quantitative real-time reverse transcription polymerase chain reaction (RT-qPCR; Figure 4). The mRNA level of proapoptotic genes (*BAX* and *CAS9*) in the vitrified group was significantly higher, while that of the anti-apoptotic gene (*BCL2*) was significantly lower compared to the control. Astaxanthin significantly upregulated the expression of *BCL2* mRNA compared to the vitrified group but significantly downregulated the expression of *CAS9* mRNA compared to both vitrified and control groups. Expression of the proapoptotic *BAX* gene in the AX-supplemented oocytes was not significant compared to either vitrified or control groups. No differences were observed in the *CAS3* gene among all tested groups.

Among the oxidative stress-related genes, mRNA expression of *CAT* and *GPX4* genes was significantly decreased in the vitrified group compared to control. AX significantly upregulated *GPX4* mRNA and only slightly upregulated *CAT* mRNA gene expression compared to the control level. *SOD2* gene expression did not differ among all tested groups.

Expression of development-related *CDX2* gene was significantly lower in the vitrified group compared to control, while AX significantly upregulated mRNA expression. AX significantly upregulated *GJB5* mRNA gene expression even compared to control.

## 4. Discussion

Vitrification can induce damage to the endogenous antioxidant systems, which can be partially repaired by the exogenous addition of antioxidants [32]. It has also been reported that additional incubation of oocytes after warming can lead not only to the restoration of the meiotic spindle [33], but also to the restoration of mitochondrial function or to a reduction in ROS formation [32,34] compared to the oocytes immediately after warming. In our previous study [31], we observed that glutathione (5 mM), added during post-warm recovery of vitrified/warmed bovine oocytes, suppressed the ROS formation in oocytes and induced faster blastocyst formation with reduced incidence of apoptotic cells.

The aim of this study was to evaluate the effect of astaxanthin during the short recovery period of oocytes post-warming (3 h), as an alternative to most studies, where astaxanthin was present during the whole period of in vitro maturation or culture.

Due to the limited amount of biological material, we tested only one concentration of astaxanthin (2.5 µM), based on previous studies [26,27]. Our obtained results confirmed the antioxidant properties of astaxanthin on vitrified/warmed oocytes. AX significantly reduced the ROS level in vitrified oocytes already after a short recovery period. A positive effect of AX given at lower concentrations (12.5 and 25 nM) on the suppression of ROS production was also reported in vitrified porcine oocytes [27] and mature bovine oocytes affected by heat shock [16]. The effect of AX at much higher concentrations (500 µM) on the reduction in ROS level was also reported during in vitro growth of bovine oocytes derived from early antral follicles [35]. Conversely, in blastocysts obtained from vitrified parthenogenetically activated porcine zygotes, reduced ROS formation was not observed after the addition of 1.5 µM AX during in vitro culture [27]. The effect of AX can, thus, differ depending on the concentration, the length of application, as well as on the stage at which the oocytes and/or embryos are.

A low and balanced ROS level is adequate for normal cell function, while vitrification/warming can increase ROS production in oocytes. When ROS production is increased, the scavenging is insufficient, and oxidative stress occurs. This results in a negative impact on the embryo development [36]. Increased ROS production might be associated with increased lipid peroxidation, as free radicals attack lipids containing carbon–carbon double bond(s), especially polyunsaturated fatty acids [37]. This was also confirmed by our results, showing increased lipid peroxidation in the vitrified group without AX compared to control, while the addition of AX significantly reduced lipid peroxidation in vitrified oocytes. The same observations are reported by [38], where astaxanthin reduced lipid peroxidation as well as ROS production induced by bisphenol A in mouse follicles. Astaxanthin appears to protect the cell membranes of cultured follicles from lipid peroxidation by reducing the production of malondialdehyde, a metabolite of membrane lipid peroxidation in follicles [38].

The mitochondria represents one of the most important organelles inside the oocyte, as they are the energy factory of the cell with a highly dynamic, double-membrane structure [39]. Mitochondrial homeostasis is a key factor for cell survival, and mitochondrial dysfunction leads to increased production of ROS through the redox pathways in oocytes. Mitochondrial dysfunction induced by vitrification can be alleviated after warming when damaged mitochondria are removed and de novo synthesis occurs to restore mitochondrial function [40]. Mitochondrial activity in different time intervals (0, 30, 60, and 120 min) after warming of oocytes reached the level of mitochondrial activity in fresh oocytes after 120 min [41]. We observed a similar trend after the recovery period, when astaxanthin further promoted mitochondrial activity compared to the fresh control group. In the case of lysosomes, vitrification/warming significantly increased lysosomal activity compared to the control and astaxanthin groups. This indicates that increased lysosomal activity can be associated with induced autophagy as a survival response. Similar trends in mitochondrial and lysosomal activities were observed in porcine four-cell embryos and blastocysts after astaxanthin addition [42].

We did not observe an effect of astaxanthin on the development of embryos from vitrified bovine oocytes. Vitrification significantly reduced the blastocyst rate (14.34%) compared to fresh control (29.85%), and the addition of AX at a concentration of 2.5 µM during the recovery period was not sufficient to improve the blastocyst rate (17.26%). It is more likely that effects of AX on oocyte and embryo development vary depending on the concentration used. Thus, [42] described the accelerated formation of blastocysts from porcine zygotes vitrified after parthenogenesis as well as higher blastocyst rate at the concentration of 1.5 µM AX. Increased blastocyst rate after astaxanthin addition was also obtained in aging porcine oocytes [26]. The addition of astaxanthin at 0.838 µM during in vitro maturation and culture significantly improved cleavage and blastocyst rates as well as a total cell number of bovine embryos after somatic cell nuclear transfer [43]. AX, when given at 50 µM, did not affect blastocyst rate, while its higher concentrations (100, 200, and 500 µM) had a negative effect on the development (13.5–15.3%) compared to control without AX (24.4%)**.** Conversely, when the AX concentration was reduced to 12.5 and 25 nM, authors observed an increase in the blastocyst rate in oocytes exposed to heat stress. High concentrations of AX seem to lead to an imbalance between oxidants and antioxidants, generating reductive stress in cumulus–oocyte complexes [16]. These results agree with the finding that overdosed antioxidant supplements can lead to “antioxidant stress” [5] and that high concentrations of antioxidants could act as pro-oxidants, i.e., increase oxidative stress and induce toxicity [44].

Viability assays based on fluorescent staining showed significantly lower total number of cells in blastocysts from the vitrified group, while AX addition increased proliferation up to the control level. More than 45% of blastocysts from the vitrified group have less than 80 cells compared to 22% in control and 20% in the vitrified AX-supplemented group, which indicates delayed blastocyst development after vitrification and the ability of AX to promote cell proliferation. Astaxanthin can incorporate into the cytoplasm of bovine preimplantation embryos and accumulate in mitochondria, thus increasing the mitochondrial potential in heat-shocked bovine embryos [45]. This can be related to the increased proliferation after astaxanthin addition, as mitochondria provide the energy for proper development.

Delayed development was also observed in embryos of human vitrified/warmed oocytes [46] and mouse vitrified embryos [47]. Our results of the expression of genes related to development (*GJB5*, *CDX2*) also support our observation of the delayed development in the vitrified group, where the *CDX2* gene expression was significantly lower in the vitrified group compared to control, while AX upregulated expression of the *CDX2* gene. The *CDX2* gene is involved in cell growth and division and expressed in the trophoectoderm at the blastocyst stage [48]. Embryos with low *CDX2* expression may have reduced viability [49]. In the case of *GJB5*, we observed significant increases in its expression after the AX addition compared to the control group. GJB5 (gap junction protein B5) belongs to the group of connexins that regulate cell proliferation necessary for normal embryonic development. Analysis of mouse *GJB5*-deficient trophoblast stem lines clearly indicates delayed trophoblast differentiation [50]. The increased expression of *GJB5* after astaxanthin addition may be related to the mechanism supporting cell proliferation in blastocysts.

RT-qPCR analysis of apoptosis-related genes *BCL2*, *BAX*, *CAS3*, and *CAS9* showed different expression patterns in tested groups. Pro-apoptotic and anti-apoptotic proteins from the BCL2 protein group regulate apoptosis during embryogenesis, so that the pro-apoptotic *BAX* gene stimulates apoptosis, while the *BCL2* gene suppresses apoptosis [51]. In the vitrified group, the *BAX* gene expression was significantly higher than in the control group, and conversely, the *BCL2* gene expression was significantly lower compared to the control group, indicating a higher chance of apoptosis. Astaxanthin significantly upregulated expression of *BCL2* and downregulated expression of *BAX* genes up to the control level. The ratio between *BAX* and *BCL2* gene expression is a reliable parameter for predicting the trend of embryo survival and apoptosis [52]. It has been published that astaxanthin increased the expression of the *BCL2* gene in aging porcine oocytes [26] and in mouse oocytes negatively affected with bisphenol A [38]. Expression of the initiation caspase *CAS9* gene was significantly downregulated compared to the vitrified group without AX, which indicates that AX can inhibit apoptosis through the regulation of apoptosis-related genes. However, in the case of execution caspase CAS3, no effects of vitrification or astaxanthin were observed. This corresponds to our results of the apoptotic cell occurrence, where no difference among all groups was observed. The reason why AX downregulated the expression of the *CAS9* gene without affecting the *CAS3* gene remains unclear. Previous study on porcine vitrified oocytes did not demonstrate differences in fluorescent labeling of caspase-3 [27]. In heat-shocked porcine oocytes, the addition of astaxanthin (0.419–1.656 µM) increased blastocyst ratio with no effect on total cell number and incidence of apoptotic cells [53].

Oxidative stress-related genes (*CAT*, *GPX4*, and *SOD2*) in our study showed different expression patterns in our tested groups. Expression of the *CAT* (catalase) gene was significantly lower in the vitrified group than in control group, while AX upregulated *CAT* gene expression to the control level. In the case of the *GPX4* gene, vitrification significantly downregulated its expression, while astaxanthin upregulated its expression up to the control level, suggesting a protective function of AX against oxidative stress caused by vitrification. The *GPX4* (glutathione peroxidase 4) gene inhibited apoptosis and lipid peroxidation in developing embryos [54]. Lack of *GPX4* gene expression leads to increased DNA fragmentation and subsequently to abnormal embryo development [55]. Expression of the *SOD2* (superoxide dismutase-2) gene did not differ among all tested groups.

The effect of vitrification on the actin cytoskeleton quality has been previously documented [56,57]. Actin is an abundant protein with well-established roles in processes ranging from cell migration to membrane transport [58]. Warmed embryos, which rehydrate, normally displayed progressive repolymerization of cytoplasmic microtubules and microfilaments [59]. Ref. [60] observed an abnormal pattern of F-actin organization in bovine vitrified/warmed oocytes cultured one hour after warming. We did not find any differences in the actin quality of blastocysts between vitrified and control groups, while the addition of astaxanthin slightly increased the ratio of blastocysts with intact actin (82.76%) compared to the embryos from the vitrified (70.83%) and control (70.73%) groups. Similarly, [61] did not observe notable differences in the actin filament structure between vitrified/warmed horse embryos after 24 h culture and fresh embryos. According to the results of the above-mentioned studies, it is obvious that cryopreservation causes changes in the organization of actin cytoskeleton, which may be reversible after post-warm recovery and can be improved by the addition of antioxidants.

Our results demonstrate an improvement in blastocyst quality after the addition of astaxanthin compared to the embryos from the vitrified group without its supplementation. Although the exposure time and concentration of AX were apparently not optimal enough for improvement of embryo development, AX mitigated the adverse effect of vitrification, which was confirmed by improved oocyte quality.

## 5. Conclusions

Astaxanthin mitigated oxidative stress in warmed bovine oocytes by suppressing ROS formation, lipid peroxidation, and increased mitochondrial activity. Although AX addition at 2.5 µM during post-warm recovery of oocytes did not significantly increase the blastocyst yield, it significantly increased cell proliferation in blastocysts from vitrified oocytes. These blastocysts exhibit better quality than slowly developing blastocysts from vitrified oocytes without astaxanthin addition. The recovery period, tested in this study, confirmed its justification for carrying out reparative processes taking place in oocytes after warming, which were further stimulated by the presence of astaxanthin.

## Figures and Tables

**Figure 1 antioxidants-13-00556-f001:**
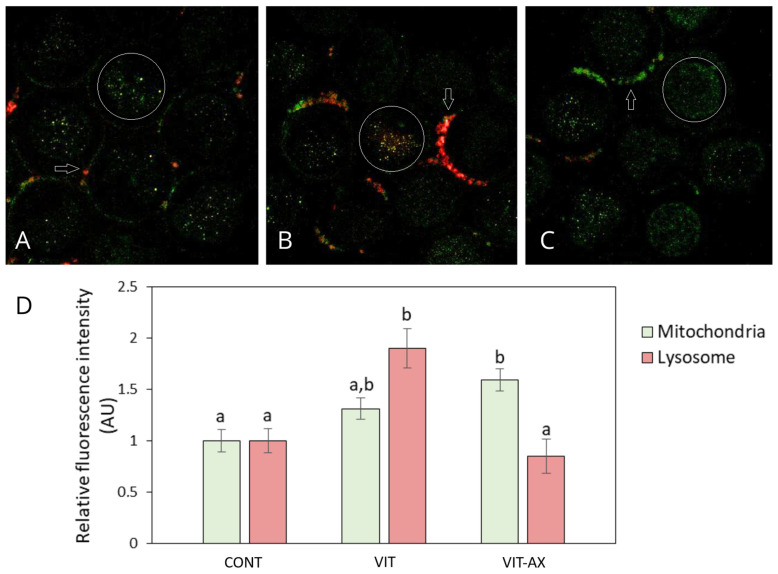
Effect of AX on mitochondrial and lysosomal status in vitrified bovine oocytes. Representative images of oocytes stained for mitochondria (MitoTracker^®^ Green, green) and lysosomes (LysoTracker™ Deep Red, red) in control (**A**), VIT (**B**), and VIT-AX (**C**) groups. Highlighted white circles indicate the oocyte ooplasm in which relative fluorescence was measured. White arrows indicate cumulus cells. Mitochondrial and lysosomal activities were evaluated based on relative fluorescence intensity in control, VIT, and VIT-AX groups (**D**). Relative fluorescence intensity for the control group was set arbitrarily at 1. Values are the mean ± SEM. Different superscripts indicate significant differences at *p* < 0.05.

**Figure 2 antioxidants-13-00556-f002:**
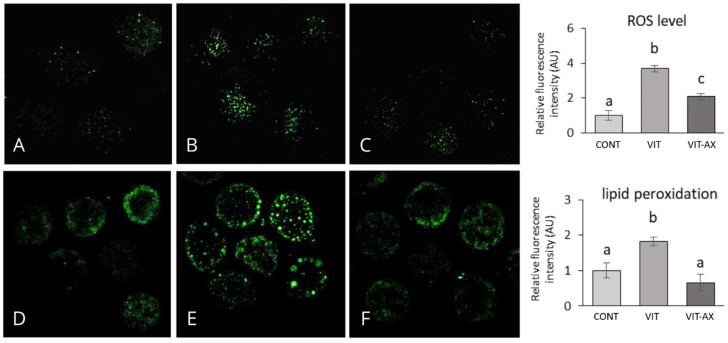
Effect of AX on ROS production and lipid peroxidation in bovine vitrified oocytes. Representative images of ROS and lipid peroxidation staining in control (**A**,**D**), vitrified oocytes without AX (**B**,**E**), and vitrified oocytes with AX (**C**,**F**). Relative fluorescence intensity for the control group was set arbitrarily at 1. Values are presented as the mean ± SEM. Different superscripts indicate significant differences at *p* < 0.05.

**Figure 3 antioxidants-13-00556-f003:**
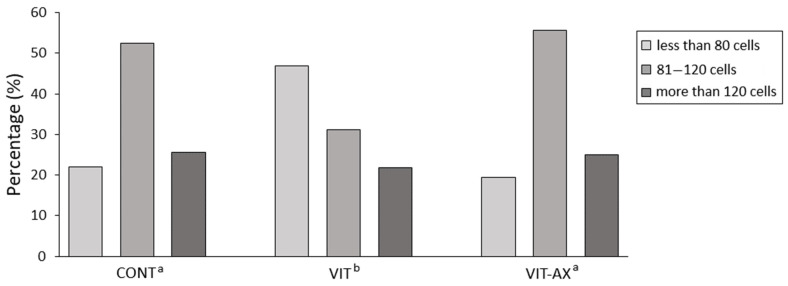
Distribution of blastocysts according to the cell number in tested groups. Different superscripts indicate significant difference between groups at *p* < 0.05.

**Figure 4 antioxidants-13-00556-f004:**
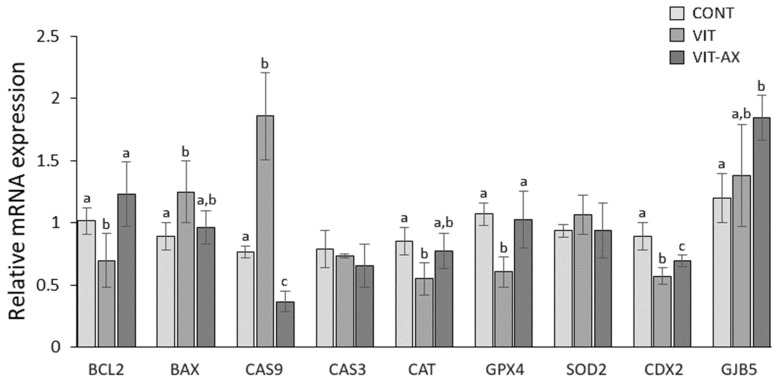
Effect of vitrification and AX on relative mRNA expression of apoptosis-, oxidative stress-, and development-related genes in blastocysts. Different superscripts (a–c) indicate a significant difference between groups (*p* < 0.05).

**Table 1 antioxidants-13-00556-t001:** Primers used for gene expression analysis.

Gene	Primer Sequences	Product Size (bp)	Tac (°C)	GenBank Accession Number
*BCL2*	F: CCT GTT TGA TTT CTC CTG R: ATA TTA TTT CTG CTG CTT CT	157	53	NM_001166486
*BAX*	F: TGA AGC GCA TCG GAG ATG AAT R: CCT TGA GCA CCA GTT TGC TG	183	62	NM_173894
*CAS9*	F: GCT AAT AAG ACT CTC ATC AA R: AAT AAC TAA CCA CCA GAA G	112	56	NM_001205504
*CAS3*	F: ACT GAT AAG AGC GTG AAC T R: CCA ACT GAC TGA CTG ACT	100	56	NM_001077840.1
*CAT*	F: TCG CTG GAT GGA AGA TTC R: CCC ACA GGA AAG TAG GAT T	122	58	NM_001035386
*GPX4*	F: GGA GCC AGG GAG TAA TGC AG R: GAC CAT ACC GCT TCA CCA CA	221	55	NM_001346431
*SOD2*	F: GTG ATC AAC TGG GAG AAT R: AAG CCA CAC TCA GAA ACA CT	160	56	NM_201527
*CDX2*	F: TCA CTC ACT AAT GTT TAC R: AAT CTA GGA GAA TGT CAT	110	45	NM_001206299
*GJB5*	F: ACG TGG TGG ACT GCT TCA TC R: GAG GAG ATC GCC CTG TTT GG	221	55	NM_001205907
*H2AFZ*	F: AGG ACG ACC AGT CAT GGA CGT GTG R: CCA CCA CCA GCA ATT GTA GCC TTG	209	57	NM_002106

F—primer forward; R—primer reverse.

**Table 2 antioxidants-13-00556-t002:** Effect of vitrification and AX on embryo development, cell number, and apoptosis.

Groups	Oocytes for IVF, n	Cleavage Rate, n (%)	Blastocyst Rate, n (%)	Total Cell Number, n	TUNEL- Index, %
CONT	201	133 (66.17) ^a^	60 (29.85) ^a^	103.80 ± 2.81 ^a^	10.15 ± 0.65 ^a^
VIT	258	150 (58.14) ^a,b^	37 (14.34) ^b^	94.03 ± 5.08 ^b^	12.16 ± 1.10 ^a^
VIT-AX	226	124 (54.87) ^b^	39 (17.26) ^b^	105.28 ± 4.45 ^a^	11.95 ± 1.19 ^a^

VIT—vitrified oocytes without AX; VIT-AX—vitrified oocytes with 2.5 µM of astaxanthin. Different superscripts in the column indicate significant differences at *p* < 0.05.

**Table 3 antioxidants-13-00556-t003:** Effect of vitrification and astaxanthin on quality of actin cytoskeleton in blastocysts.

Groups	Blastocysts, n	Bl with Intact Actin, n (%)	Bl with Damaged Actin, n (%)
CONT ^a^	41	29 (70.73)	10 (29.27)
VIT ^a^	24	17 (70.83)	7 (29.17)
VIT-AX ^a^	29	24 (82.76)	5 (17.24)

VIT—vitrified oocytes without AX; VIT-AX—vitrified oocytes with AX; Bl—blastocyst. ^a^ Differences between groups were not significant.

## Data Availability

The data are contained within this article.

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
