# Peer review of "Astaxanthin Added during Post-Warm Recovery Mitigated Oxidative Stress in Bovine Vitrified Oocytes and Improved Quality of Resulting Blastocysts"

_antioxidants, 2024, doi:10.3390/antiox13050556_

Round 1

Reviewer 1 Report

The authors investigated the effects of astaxanthin addition to culture medium for vitrified and thawed bovine oocytes, and concluded that astaxanthin improve quality of blastocysts developed from the oocytes. The study provides detailed examinations and its interpretation seems generally appropriate, however there are some points to be elucidated for publication.

Line 102-103: Why was astaxanthin not applied during vitrification, even though the aim of this study was improving the quality of oocytes after vitrification/warming?

Line 158-161: The authors should show pictures of actin staining and difference between intact and damaged filaments with classificatory criterion.

Line 228: What were the criteria for selection for IVF?

Line 229: “which represents 72.80 % viability of oocytes prior to vitrification” is confusing. Was IVF performed before or after vitrification? If it is after vitrification, “prior to” must be changed into “without” or others.

Line 261: “number” should be “rate”.

Line 278-279: This statement does not match the graph (Fig. 4).

Line 294-312: These statements are repetitions of the introduction section and are not necessary for the discussion section.

Line 326-327, 358-361: If so, how and why the authors decided the only one concentration of astaxanthin in this study?

Line 338-340: The authors need to show evidence or reference(s) for this explanation.

Line 397-398: The authors need to analyze what will occur when astaxanthin is added to non-vitrified oocyte. Was it possible that the supporting cell proliferation was enhanced while the total cell number decreased in blastocysts?

Line 421: It is unclear that “our study” indicates their current study or their previous study.

Line 442: The current results do not support this explanation (rather the opposite) and are therefore inappropriate.

Line 445: It is unclear what blastocyst quality indicates (probably only total cell number and actin filament in this study. Quality of blastocyst, in a direct sense, would be the development ability of embryo. Thus, the authors need to explain the relationship between development ability of embryo and total cell number and actin filament in blastocyst.

Author Response

Dear Reviewer,

Thank you very much for taking the time to review this manuscript. Please find the detailed responses below.

Line 102-103: Why was astaxanthin not applied during vitrification, even though the aim of this study was improving the quality of oocytes after vitrification/warming?

Response: Astaxanthin was not applied during vitrification, because the equilibration process last only 12 minutes and exposure to vitrification solution only 25 seconds. This period is too short to show some effect of the astaxanthin. Moreover, for the survival of oocytes after cryopreservation, the warming process is more critical than vitrification process itself.

Line 158-161: The authors should show pictures of actin staining and difference between intact and damaged filaments with classificatory criterion.

Response: Thank you for suggestion. Actin classification criteria were specified and illustrated in our previous study. A reference to this study has been added in the text.

Line 228: What were the criteria for selection for IVF?

Response: The criteria for selection of oocytes for IVF were oocytes with homogeneous ooplasm with several layers of cumulus cells. Oocytes with shrunken or extra light ooplasm, completely denuded oocytes or oocytes with sign of degeneration were not suitable for IVF and were therefore discarded.

Line 229: “which represents 72.80 % viability of oocytes prior to vitrification” is confusing. Was IVF performed before or after vitrification? If it is after vitrification, “prior to” must be changed into “without” or others.

Response: The IVF was performed after vitrification/warming in VIT and VIT-AX group. From total of vitrified oocytes, 72.80 % were selected for IVF. Phrase “prior to” was, therefore, changed into “after”.

Line 261: “number” should be “rate”.

Response: Phrase was corrected.

Line 278-279: This statement does not match the graph (Fig. 4).

Response: Thank you for the warning, the text has been edited.

Line 294-312: These statements are repetitions of the introduction section and are not necessary for the discussion section.

Response: The text has been edited and repetitions of the introduction section were removed.

Line 326-327, 358-361: If so, how and why the authors decided the only one concentration of astaxanthin in this study?

Response:  For the limited amount of biological material, we chose only one concentration. The concentration was chosen based on mentioned studies (Jia et al., 2020 and Xiang et al., 2021) that obtained the best effect of astaxanthin at 2.5 µM when testing the different concentrations.

Line 338-340: The authors need to show evidence or reference(s) for this explanation.

Response: The reference was added.

Line 397-398: The authors need to analyze what will occur when astaxanthin is added to non-vitrified oocyte. Was it possible that the supporting cell proliferation was enhanced while the total cell number decreased in blastocysts?

Response: Our primary aim was to observe the effect of astaxanthin on vitrified oocytes. For this reason, we did not test its effect on fresh oocytes. Based on the available literature, it is possible that astaxanthin would increase the development to blastocyst stage and increased total cell number.

Line 421: It is unclear that “our study” indicates their current study or their previous study.

Response: Expression “our study” in this sentence indicates the results of current study. The sentence has been edited.

Line 442: The current results do not support this explanation (rather the opposite) and are therefore inappropriate.

Response: We are afraid, that we do not comprehend this reviewer comment. Could you please clarify what you mean concretely.

Line 445: It is unclear what blastocyst quality indicates (probably only total cell number and actin filament in this study. Quality of blastocyst, in a direct sense, would be the development ability of embryo. Thus, the authors need to explain the relationship between development ability of embryo and total cell number and actin filament in blastocyst.

Response: Blastocyst quality in our results covers, besides the total cell number and actin cytoskeleton status, also expression levels of important genes, which we analyzed in this study. Developmental ability (cleavage and blastocyst rates) may be greatly affected also by an original quality of oocytes flushed from the slaughterhoused ovaries. However, for qualitative markers we took only embryos which reached the blastocyst stage, i.e. in this case a factor of initial oocyte quality was excluded.

Reviewer 2 Report

In this study, the researchers observed that astaxanthin, acting as a natural antioxidant, can alleviate the negative effects of vitrification and enhance the quality of blastocysts. Specifically, the appropriate concentration of astaxanthin added to the maturation medium during a 3-hour recovery period following vitrification can effectively restore cell proliferation, as well as mitochondrial and lysosomal activity, by reducing the formation of reactive oxygen species (ROS). However, there are several concerns that need to be addressed.

  1. In the "Introduction" section, it should be emphasized that the primary function of cryopreservation is to preserve and transport gametes from genetically valuable individuals, with the secondary function being the potential increase in their offspring numbers.

  2. The fluorescent images of oocytes should include their corresponding bright-field images or at least delineate the range of oocytes, accompanied by annotations explaining the significance of each color depicted.

  3. Despite the beneficial effects observed, it's noteworthy that astaxanthin did not lead to an increase in blastocyst formation rates. Therefore, additional indices should be explored to corroborate the claim of improved blastocyst quality, such as the expression of totipotent markers like OCT4, among others.

Author Response

Dear Reviewer,

Thank you very much for taking the time to review this manuscript. Please find the detailed responses below.

  1. Thank you for your suggestion. The sentence has been rewritten.
  2. The figure was edited. Since we don´t have representative corresponding bright-field images, we have marked the measured area of fluorescence and edited the figure description.
  3. We agree with your comment, that the totipotency marker -OCT4 would be a proper parameter to consider about blastocyst quality. Unfortunately, in the actual situation we cannot perform this analysis from technical reasons (those samples are not available anymore). Nevertheless, we will consider about this assay in our further experiments.